# UNSUPERVISED EVENT OUTLIER DETECTION IN CONTINUOUS TIME

## ABSTRACT

Event sequence data record the occurrence times of various events. Event sequence forecasting based on temporal point processes (TPPs) has been extensively studied, but outlier or anomaly detection, especially *unsupervised* detection of abnormal events, is still underexplored. In this work, we develop, to the best our knowledge, the first unsupervised outlier detection approach to detecting abnormal events. Our novel unsupervised outlier detection framework is based on ideas from generative adversarial networks (GANs) and reinforcement learning (RL). We try to train a "generator" that corrects outliers in the data with the help of a "discriminator" that learns to discriminate the corrected data from the real data, which may contain outliers. Different from typical GAN-based outlier detection approaches, our method employs the *generator* to detect outliers in an *online* manner. The experimental results show that our method can detect event outliers more accurately than the state-of-the-art approaches.

## 1 INTRODUCTION

Event sequence data are records of the occurrences of different events in continuous time, such as buy-sell orders in a financial market, natural disasters in a country, or user actions when using an app. They can be represented as individual points on a time line, with the location of the points indicating the time of the event occurrences.

For event sequence data, forecasting and latent structure inference have been the focuses of most previous research. Methods based on Gaussian processes (e.g., (Rao & Teh, 2011; Lloyd et al., 2015; 2016; Ding et al., 2018; Liu & Hauskrecht, 2019)), Hawkes processes (e.g., (Zhou et al., 2013; Lee et al., 2016; Xu et al., 2016; Wang et al., 2016; Kim, 2018)), and more recently deep neural networks (e.g., (Du et al., 2016; Mei & Eisner, 2017; Xiao et al., 2017b; Omi et al., 2019; Zhang et al., 2020; Zuo et al., 2020; Xue et al., 2022)), have been widely proposed and evaluated. These methods try to either predict the times, and in some cases also labels, of the future events given the observed history, or infer the latent structure driving the event occurrences. Temporal point processes (TPPs) are the common probabilistic model shared by almost all these methods for modeling discrete events in continuous time.

In contrast to forecasting, sometimes the event occurrences themselves can be unexpected, and it would be valuable to detect those cases. For example, an unexpected credit card transaction may indicate fraud, which can cause significant financial losses for the card holder and the financial institute. An unexpected medicine dosage in a healthcare environment may cause life-threatening consequences to the patient. On the other hand, detecting unexpected absence of events can also be valuable. For example, if a credit card user is expected to pay for a bill but the transaction did not occur in time, catching it sooner than later and alerting the user can help them avoid a late payment and associated penalty.

Recently Liu & Hauskrecht (2021) defined two types of event outliers in continuous-time event sequences corresponding to the aforementioned unexpected occurrences and absence of events. They proposed semi-supervised outlier detection methods, assuming access to clean data without any outliers to train a model. Although this is a common assumption (semi-supervised) in the literature, unsupervised methods without this assumption would have more practical value, since in reality it is usually hard to assume that data are clean without checking or preprocessing.

Inspired by Generative Adversarial Networks (GANs) (Goodfellow et al., 2014), we propose to solve this problem by modeling a "generator" that tries to find and correct outliers and a "discriminator" that tries to distinguish the "corrected" data from the real data that can be either normal or abnormal. For either commission (unexpected occurrences) or omission (unexpected absence) outliers, we define an individual pair of generator and discriminator. The key insight is that, for each outlier type, a generator can either correctly remove the outliers in the real data or incorrectly add new outliers of the other type. If the former is the case, it will be very difficult for the discriminator to separate the "corrected" data from the normal samples in the real data, which, by definition, constitute the majority of the data. Meanwhile, if the latter is the case, it will be relatively easy for the discriminator to separate. This intrinsic contrast between these two cases will be the source of feedback for both the generators and the discriminators to learn. Once learned, the "generators" can be used in an online manner to detect outliers on unseen event sequences.

The discriminator can be trained using stochastic gradient descent as a classification problem. However, gradient descent-based optimization cannot be used for the generator because outlier correction our case is non-differentiable. There are various ways for handling the non-differentiability, such as Gumbel-softmax Jang et al. (2016), cooperative learning Lu et al. (2019); Lamprier et al. (2022) and policy gradient methods (Williams, 1992; Mnih et al., 2016; Schulman et al., 2017). We chose the latter as the approach in this paper, due to its flexibility (e.g. outlier correction in the omission case can consist of two "generators" playing a cooperative game.). Evaluating which approach is superior in our setting would be interesting but beyond the scope of this paper.

## 2 RELATED WORK

**Generative model based outlier detection** Although outlier detection methods using deep generative models are not new (e.g., (Akcay et al., 2019; Schlegl et al., 2017; Li et al., 2019; Zhu et al., 2023)), none of them detect abnormal occurrences and absence of events, and almost all of them take a semi-supervised approach, assuming availability of training data consisting of samples from a single class (usually normal data but sometimes abnormal data). Note that some authors (Schlegl et al., 2017; Li et al., 2019) just use a different terminology. In contrast, our method detects event outliers in continuous time and does it in a completely unsupervised setting, where the training event sequences contain both normal and abnormal occurrences and absence of events. Once trained, the models can be used for detecting outliers online on unseen data.

**Deep generative models for event sequences** Most deep learning models for event sequence data are based on combining deep architectures (such as RNNs (Du et al., 2016; Mei & Eisner, 2017) and attention-based (Zhang et al., 2020; Zuo et al., 2020) with TPPs, and learned by maximizing the likelihood. Different from these models, Xiao et al. (2017a) developed a Wasserstein distance for TPPs and a Wasserstein GAN to generate samples from the learned TPP. Li et al. (2018) propose to use reinforcement learning to learn a generative model utilizing inverse reinforcement learning to learn the reward from the training data. Our work is also inspired by ideas from GANs and reinforcement learning, but our goal is outlier detection instead of sequence generation, which results in different model architectures and algorithms for learning and inference.

**Outlier detection for event sequences** Liu & Hauskrecht (2021) aim to detect the same types of event outliers as in our work, but their approach requires training a point process model to learn the distribution of the data, which, in theory, would require the training data to be clean without any outliers. In contrast, our method does not require clean data and can learn directly on polluted data in a completely unsupervised fashion. Zhang et al. (2021) develop a greedy algorithm to separate exogenous events from an event sequence, which can be considered as a special case of unexpected event occurrences. Similarly Mei et al. (2019) propose a particle-smoothing algorithm for imputing missing events in event sequences. These methods focus on offline data processing and analysis, while our work focuses on unsupervised learning of models for online outlier detection. Zhu et al. (2023) propose a GAN based approach to detecting anomalous *sequences* assuming the availability of only anomalous data, while we focus on detecting anomalous occurrences and absence of *events* assuming the unlabeled data contain both normal and abnormal events. Similarly Shchur et al. (2021) propose a new statistic for goodness-of-fit testing and detecting anomalous event *sequences* instead of events.

# 3 UNSUPERVISED EVENT OUTLIER DETECTION

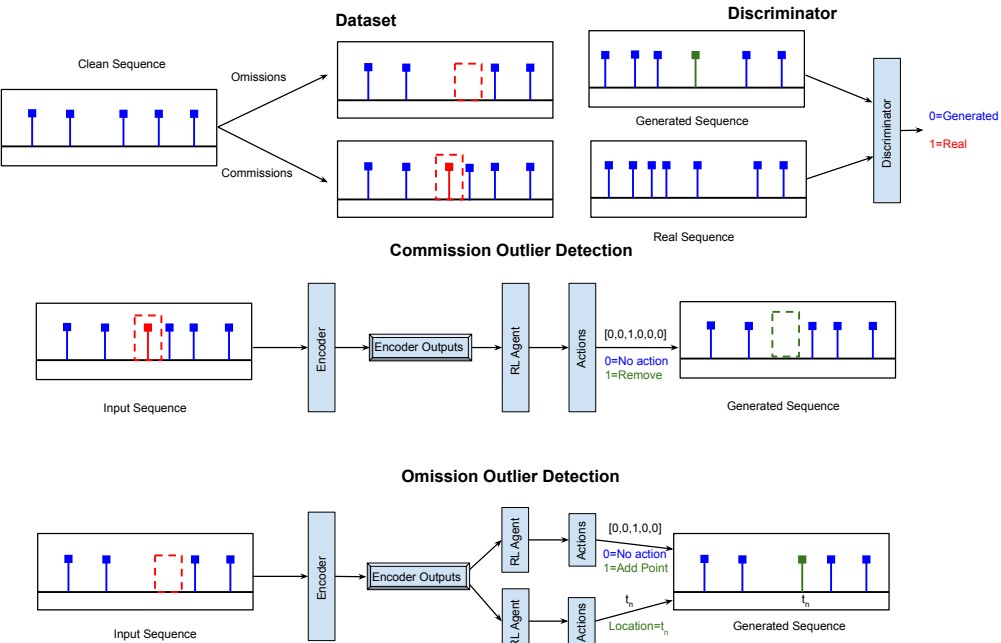

Figure 1: GAN + RL framework for unsupervised event outlier detection in continuous time.

## 3.1 PROBLEM FORMULATION

An event sequence is defined as $S = \{t_n : t_n \in \mathcal{T}\}_{n=1}^N$ where $t_n$ is the time of the occurrence of event $n$, $N$ is the total number of events, and $\mathcal{T}$ denotes the entire time domain. We assume access to a dataset $\mathcal{D} = \{S_i\}_{i=1}^I$, consists of $I$ event sequences, where some sequences might be corrupted in the form of either the addition of abnormal points or the removal of normal points, and our goal is to learn an outlier detector for each type of outliers that can be applied online to new event sequences.

**Commission outliers (unexpected occurrences of events):** Given an event point at $t_n$ in an event sequence $S$, the goal is to identify whether the event is an outlier or not, so the output would be a label, $y_n \in \{0, 1\}$, assigned to each event point.

**Omission outliers (unexpected absence of events):** For each interval $B_n = (t_{n-1}, t_n)$ between two consecutive event points $t_{n-1}$ and $t_n$ in an event sequence $S$, the goal is to decide whether there were any events that should have happened during the interval but are instead missing, so the output would be a label, $y_{B_n} \in \{0, 1\}$, assigned to each interval between two consecutive points.

In order to detect outliers in an unsupervised and online manner, we develop GAN-based approaches to training a pair of outlier detectors without any supervision. More specifically, we model a **generator that produces "corrected" sequences** given input sequences from the potentially corrupted data and a **discriminator that distinguishes between the "real" and "corrected" sequences**, for each type of outliers.

We sample sequences from the potentially corrupted data as "real" sequences. Since by definition, outliers are supposed to be relatively rare compared to normal data, we can reasonably assume that the majority of the sampled sequences are normal, and if the generator performed an incorrect "correction" on the data, e.g., removing a normal point instead of an outlier point from the sequence, then it would be easier for the corresponding discriminator to distinguish it from the sampled sequences. In the following, we describe our framework in detail.

## 3.2 ENCODER

We use an encoder to summarize the information in the past for each sequence, as the occurrence of a new event can be influenced by the events that occurred before it. Since the event times are continuous and irregular, we use an architecture, the **continuous-time LSTMs (cLSTM)** (Mei & Eisner, 2017), designed especially for continuous-time event sequences. Taking as input an event sequence $S_i$, the cLSTM outputs a latent vector of dimension, $z_n \in \mathcal{R}^D$ for each point $t_n \in S$. Thus, for each sequence of length $N_i$,

$$\{z_n\}_{n=1}^{N_i} = f_{\text{cLSTM}}(\{t_n\}_{n=1}^{N_i}).$$

We can directly use $z$'s as inputs to the generator, but since we wish to detect and correct outliers, information about every point in the past can be crucial for determining the action, so we apply the *attention* mechanism (Bahdanau et al., 2016) followed by *layer normalization* (Ba et al., 2016) to the latent outputs from the cLSTM with a causal mask to ensure that there is no information leak from the future, and therefore the learned outlier detectors can be applied online. Thus, the final outputs from the encoder are $\phi_n \in \mathcal{R}^D$,

$$\{\phi_n\}_{n=1}^{N_i} = \text{Attention}_{\text{causal}}(\{z_n\}_{n=1}^{N_i}).$$

This entire architecture, consisting of the cLSTM and attention layer, forms the encoder as shown in Figure 1.

## 3.3 GENERATOR

For our problem, each generator is modeled as a Reinforcement Learning (RL) agent that tries to identify and correct outliers in continuous-time event sequences. The input is an event sequence from the dataset, which is consistent for both the commission and omission outlier detection problems, although the architectures of the agents differ.

Each sequence $S_i$, before being fed into the RL agent, is first passed through an encoder as we described in the previous section. The final encodings $\phi$'s are treated by the RL agent as sequential states of the environment, whose goal is either to identify and remove outlier points from the input sequence or to identify and impute in unexpected blank intervals in the input sequence. Each sequence is treated as an episode, in which the agent makes decisions about making changes to the sequence. To achieve this, we use policy gradient methods, which are effective for parameterizing the optimal policy and exhibit good performance in these types of tasks (Yoon et al., 2019).

For our paper, we use the clip version of Proximal Policy Optimization (PPO) algorithm (Schulman et al., 2017), a popular RL algorithm that works well in both discrete and continuous action spaces. Moreover, PPO has been shown to perform well in a variety of tasks, including those with high-dimensional observation and action spaces. One of the advantages of PPO is that it is easy to implement and tune. Additionally, PPO has been shown to be more sample-efficient than other popular algorithms like A2C (Advantage Actor Critic) (Mnih et al., 2016) and DDPG (Deep Deterministic Policy Gradient) (Lillicrap et al., 2019).

**Commission outlier detection:** For commission outlier detection, the RL agent tries to identify and remove outlier points in the sequence. For each point $t_n$ in the sequence, the RL agent takes an action ($a_n \in \{0, 1\}$): either to keep $t_n$ ($a_n = 0$) or to remove it ($a_n = 1$). The RL agent thus learns **a policy $\pi$ that defines the probability of a point being an outlier**, and if the action sampled from the policy is 1, the point is removed from the sequence. The generated sequence consists of all the points untouched by the RL agent. In this way, by hopefully keeping only the normal points, it generates a new sequence, which is fed into a discriminator to evaluate how "real" it is.

**Omission outlier detection:** For omission outlier detection, we need a more complex architecture. This is because the input sequence that we receive may have missing points, which would act as outliers. To generate a sequence that is free of outliers, we must identify the locations of the missing points in the sequence. To achieve this, we need to (1) determine whether there is any missing point in an interval, and (2) if so, the location of the missing point. We assign each of the tasks to one of the *two agents*. Both agents take a sequence of points as input, and the first agent predicts **the probability of having a missing point in the interval between the current and previous points**.

Once the first agent identifies that there should have been a point in the interval, the second agent generates **the location of the missing point**.

The first agent operates very similarly to the commission outlier setting and outputs the probability of a point being missing in the previous interval. The policy of the first agent $\pi_1$ outputs a probability of whether there was a missing point in the previous interval with a discrete action space. The second agent has a continuous action space and tries to output the exact location $t_n$ of the missing point as shown in Figure 1. Thus we let its policy $\pi_2$ output a continuous value in $(0, 1)$ representing the relative location in the interval between the current and previous points. Once we have generated the new sequence in this manner, we feed it to the discriminator to evaluate how close the generated sequence is to a real sequence.

For omission detection, we can still use PPO as this algorithm can easily incorporate both discrete and continuous action space, which is what we need. It is worth noting that the actions of the second agent depend on the decisions of the first. As a result, during early training, the poor choices made by the first agent can negatively affect the second agent as well. To prevent this from happening, the second agent starts off at a much slower learning rate compared to the first agent. The learning rate of the second agent is increased gradually as the first agent learns and becomes more accurate in predicting missing points in the sequences.

The main generator objective for both types of outlier detection would thus be:

$$L_G(\theta_g) = \mathbb{E}[L^{\text{CLIP}}(\theta_g) - c_1 L^{\text{VF}}(\theta_g) + c_2 S[\pi_{\theta_g}](\phi_n)] \tag{1}$$

where

$$L^{\text{CLIP}}(\theta_g) = \mathbb{E}[\min(\frac{\pi_{\theta_g}}{\pi_{\theta_g}^{\text{old}}}\hat{A}_n(r_n), \text{clip}(\frac{\pi_{\theta_g}}{\pi_{\theta_g}^{\text{old}}}, 1 - \epsilon, 1 + \epsilon)\hat{A}_n(r_n)]$$

$$L^{\text{VF}} = \text{MSE}(V_{\theta_g}, V^{\text{target}}(r_n))$$

Here, the policies are parameterized by $\theta_g$ and the reward $r_n$ is obtained from the discriminator. The discriminator outputs a reward at the completion of the sequence so only the terminal reward is non-zero. For the omission problem, there are two policies, each parameterized by its own $\theta_g$. PPO works by evaluating an older version of the policy($\pi_{\theta_g}^{\text{old}}$) to make sure the newer policy does not deviate much, and this is handled by a clipping loss where the ratios between the new and older policies are constrained with a clipping hyper-parameter $\epsilon$. $\hat{A}_n$ represents the advantage function at time-step $n$ which was used in (Schulman et al., 2017; Mnih et al., 2016).The advantage function represents the advantage of taking a particular action in a particular state and is defined by the difference between the action values and state values. $c_1$ and $c_2$ are hyper-parameters that weigh the different surrogate losses and S is the entropy of the probability distribution $\pi_{\theta_g}$ at state $\phi_n$. This entropy bonus is meant to encourage exploration and prevent the agent being stuck in sub-optimal policies. $L^{\text{VF}}$ is the TD error loss (Sutton, 1988), which is the mean squared error between the current value function and the bootstrapped target, $V^{\text{target}}$.

## 3.4 DISCRIMINATOR

The goal of the discriminator is to distinguish between the generated sequences, $S_j^g$, obtained from the RL agent, and the real sequences, $S_i$, sampled from the dataset $\mathcal{D}$. As mentioned earlier, since the proportion of corrupted sequences in the dataset should be low, the majority of the samples are clean sequences. The discriminator tries to **determine whether a given sequence is "real" or "generated"** using a non-linear classifier, parameterized by $\theta_d$ that outputs the probability of the sequence being real ($p_{\theta_d}$). To prevent the discriminator from dominating over the RL agent, we also add spectral normalization (Miyato et al., 2018) to each layer of the classifier model. The discriminator has the exact same architecture as the generator except without the self-attention and layer normalization. However, no weights between them are shared. Since the discriminator and the generator perform opposing tasks, sharing the encoder would make them harder to train. The input to the classifier is the mean of the cLSTM outputs ($z$'s) from a particular sequence, and the target is 0 or 1 if the sequence is generated or sampled from the dataset respectively, $Y \in \{0, 1\}$. The discriminator loss is thus a simple cross-entropy loss that tries to identify whether the input sequence is real or generated:

$$L_D(\theta_d) = \mathbb{E}[-Y \log p_{\theta_d}(S) - (1 - Y) \log(1 - p_{\theta_d}(S))]. \tag{2}$$

## 3.5 ALGORITHM

Algorithm 1 describes how the generator and discriminator are trained and what the inputs and outputs of the models look like. One additional thing to note is that, while each individual sequence is treated as an episode for the RL agent, the generator is trained only when an episode has been completed i.e. we always wait for the sequence to get completed before updating the generator. The same is true for the discriminator because it would need complete sequences as inputs anyway. Another crucial thing to keep in mind is that although this entire network can be trained end-to-end, we opt for an *iterative* training regime in the spirit of alternating gradient descent in the GAN literature (Mescheder et al., 2018). The difference is that our generator is trained by PPO instead of gradient descent. That means for the first few episodes only the discriminator is trained, and then only the generator, and so on. This is to ensure that the training dynamics are sufficiently smooth and that both the discriminator and the generator can learn gradually. An intuition on the training dynamics and more details on training is in Appendix A.1.

---

**Algorithm 1** GAN-RL

---

**Require:** Unlabeled Event Sequences $\mathcal{D} = \{S_i\}_{i=1}^I$, `Generator` ($\theta_g$), `Discriminator` ($\theta_d$), Update Frequency $F$, Number of Episodes $K$
 1: **for** $k = 0 \ldots K - 1$ **do**
 2:     Sample $S_j = \{t_n\}_{n=1}^{N_j} \sim \mathcal{D}$
 3:     $\{\phi_n^j\}_{n=1}^{N_j} = \texttt{Encoder}_{\theta_g}(S_j)$
 4:     **for** $n = 1 \ldots N_j$ **do**
 5:         $a_n \leftarrow \pi_{\theta_g}(\phi_n^j)$
 6:         $r_n = 0$
 7:     **end for**
 8:     $S_j^g = \texttt{generate}(S_j, \{a_n\}_{n=1}^{N_j})$                                   ▷ Add or remove points
 9:     **if** $k \mod F < F/2$ **then**
10:         Sample $S_i \sim \mathcal{D}$ and compute $p_{\theta_d}(S_i)$ and $p_{\theta_d}(S_j^g)$
11:         Update $\theta_d$ using $L_D$ (Eq. 2)
12:     **else**
13:         Compute $r_{N_j} = p_{\theta_d}(S_j^g)$
14:         Update $\theta_g$ using $L_G$ (Eq. 1)
15:     **end if**
16: **end for**

---

## 4 EXPERIMENTS

In the experiments, we study, despite only having access to unlabeled and potentially corrupted data, whether GAN-RL can still learn outlier detectors without supervision and beats the state-of-the-art approach originally designed for semi-supervised settings (Section 4.1). Since the performance of GAN-RL can depend on the level of corruption in the dataset, we also analyse its ability to learn from datasets with varying levels of corruption (Section 4.3). In Section 4.4 we perform an ablation study regarding the choice of adding an attention layer in the encoder.

**Datasets** To assess the performance, we use four datasets: two synthetic datasets and two real-world datasets. The synthetic datasets were generated by defining intensity functions for clean data. These intensity functions correspond to either an inhomogeneous **Poisson** process or a **Hawkes** process. Additionally, we include two real-world datasets. **MIMIC** is a dataset used in prior work in event sequence modeling, such as (Du et al., 2016; Mei & Eisner, 2017). It records the admission times of patients in an Intensive Care Unit over a period of 7 years. **Taxi** (Whong, 2014) tracks taxi pick-up and drop-off events in the New York City, and it was used in (Xue et al., 2022).

For all the datasets, we also define a parameter $\beta$, controlling the percentage of clean sequences in the dataset. If $\beta$ is 0.7, it means that 70% of the sequences are clean. For all our experiments, we use a $\beta$ of 0.8 unless otherwise mentioned. Commission outliers are generated by sampling from a Poisson process with a constant intensity function. For omission outliers, normal events in

the sequences are randomly removed. The specific intensity functions (Appendix B.1) and outlier generation mechanism (Appendix B.2) can be found in the Appendix.

**Baseline** As unsupervised event outlier detection without access to clean data has not been previously explored, we adapt the state-of-the-art approach for semi-supervised event outlier detection, PPOD, as our baseline (Liu & Hauskrecht, 2021), which has demonstrated strong performance when clean data is available. It also leverages the cLSTM architecture but is trained with a negative log-likelihood loss (Mei & Eisner, 2017) to generate the intensity functions used for scoring points and intervals in the sequences for outliers.

## 4.1    PERFORMANCE OF GAN-RL

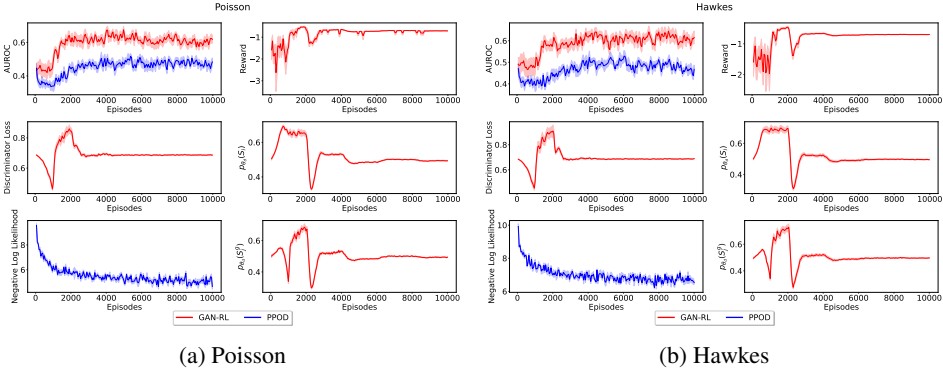

Figure 2: Training performance of our algorithm for commission outliers on synthetic datasets. The curves are plotted across 10 independent runs. The shaded regions represent the standard errors.

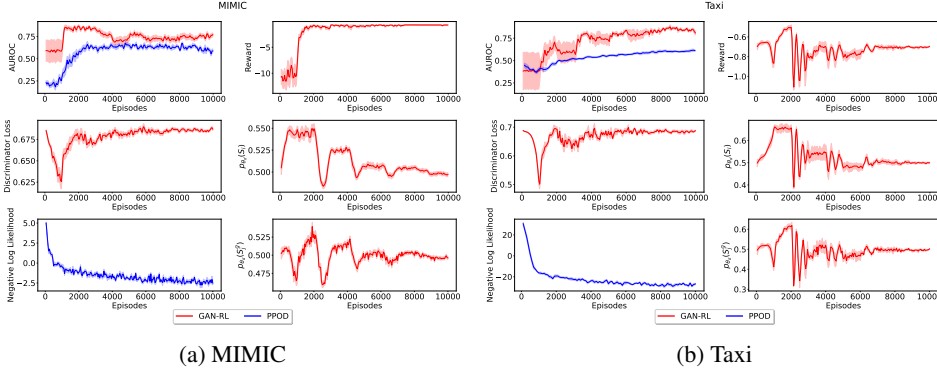

Figure 3: Training performance of our algorithm for commission outliers on real datasets. The curves are plotted across 10 independent runs. The shaded regions represent the standard errors.

**Commission Outliers:** In Figure 2 and  3, we present a comprehensive analysis of our RL agents' performance over a series of 10,000 episodes, each representing a complete sequence. For the synthetic datasets, these sequences are chosen randomly from a pool of 1000 generated sequences for training. Meanwhile, for MIMIC and Taxi, we sample from the same training data splits as used in previous work. To gauge the effectiveness of our method against the baseline method, we employ AUROC (Area Under the Receiver Operating Characteristic Curve) scores, computed based on the ground-truth labels not accessible by any of the methods during training. This metric provides valuable insights into the RL agents' ability to distinguish outliers from the norm.

Over 10 independent runs, our results consistently demonstrate that our RL agents excel at outlier detection when compared to the baseline method. The baseline barely does better than random, and we believe that this is mainly because the outliers in the data cause confusion to its underlying model attempting but not being able to learn the true intensity function. GAN-RL has a superior performance across all the datasets.

Another interesting observation worth mentioning are the probability that the real sequence is classified correctly $p_{\theta_d}(S_i)$ and the probability that the *generated* is classified as real $p_{\theta_d}(S_j^g)$ . Both these probabilities seem to be around $0.5$ which suggests that the RL agent has done a great job in generating real-looking data by correcting outliers.

**Omission Outliers:** Figure 4 & 5 show the performance of GAN-RL in comparison to the baseline. We notice similar performance improvements in this setting as well for 3 out of the 4 datasets.

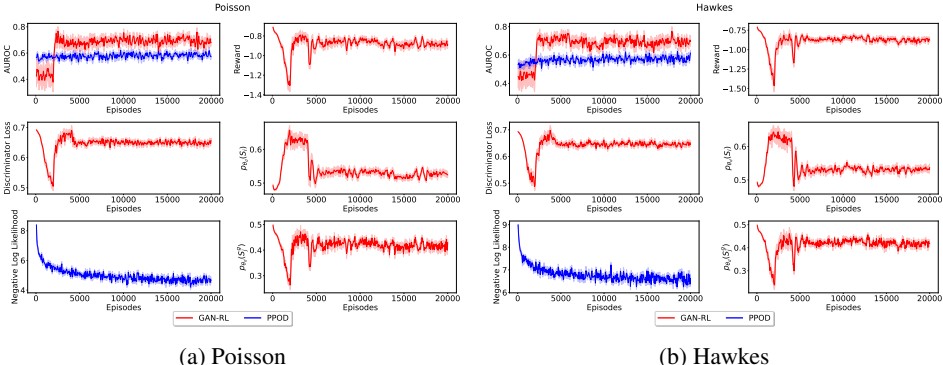

(a) Poisson                                        (b) Hawkes

Figure 4: Training performance of our algorithm for omission outliers on synthetic datasets. The curves are plotted across 10 independent runs. The shaded regions represent the standard errors.

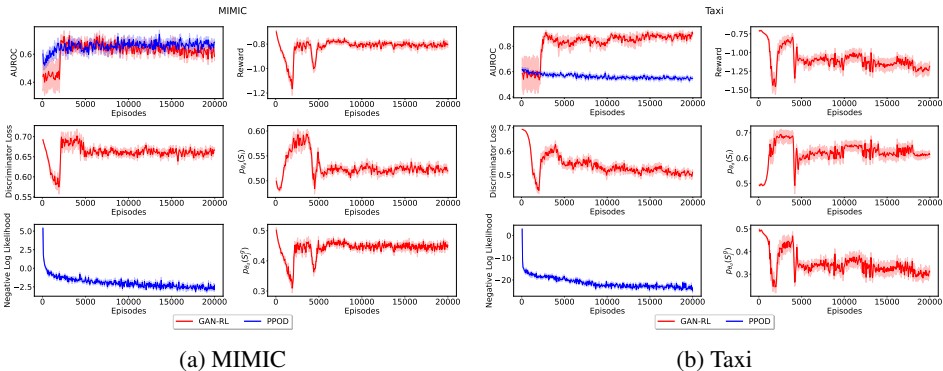

(a) MIMIC                                        (b) Taxi

Figure 5: Training performance of our algorithm for omission outliers on real datasets. The curves are plotted across 10 independent runs. The shaded regions represent the standard errors.

## 4.2 RESULTS ON TEST DATA

We also highlight test performance across all the datasets across 10 seeds in Table 1 for both commission and omission outliers. These are computed over 100 test sequences unseen during training. For the real-world datasets, we simply use the test split in the dataset, and for the synthetic datasets, we generate new sequences using seeds different from the ones used in training.

## 4.3 SENSITIVITY TO THE AMOUNT OF CORRUPTION IN THE DATA

The unsupervised nature of the problem setting necessitates the presence of *some* normal sequences (or sequence segments if we split complete sequences into shorter segments) in the dataset, and this usually should not be an issue as the majority of the data should be clean by definition of outliers. However, if that is not the case we expect GAN-RL to fail because the discriminator would perceive noisy sequences as real. To study the effect of the "cleanness" of the datasets, we plot in Figure 6 the average performance on 100 test sequences. The $X$-axis is the parameter $\beta$ and increasing $\beta$ means fewer corrupted sequences. As $\beta$ increases, we notice an improvement in the performance of

Table 1: Evaluation on test dataset for commission outliers (left) and omission outliers (right)

| Dataset | PPOD | GAN-RL |
|---------|------|--------|
| Poisson | $0.55 \pm 0.02$ | $0.631 \pm 0.03$ |
| Hawkes | $0.512 \pm 0.01$ | $0.610 \pm 0.03$ |
| MIMIC | $0.583 \pm 0.02$ | $0.778 \pm 0.12$ |
| Taxi | $0.548 \pm 0.01$ | $0.647 \pm 0.03$ |

| Dataset | PPOD | GAN-RL |
|---------|------|--------|
| Poisson | $0.580 \pm 0.07$ | $0.653 \pm 0.06$ |
| Hawkes | $0.594 \pm 0.09$ | $0.639 \pm 0.04$ |
| MIMIC | $0.554 \pm 0.06$ | $0.563 \pm 0.05$ |
| Taxi | $0.541 \pm 0.03$ | $0.689 \pm 0.07$ |

Table 2: Comparison of asymptotic training performance with and without attention.

| Dataset | GAN-RL w/o Attention | GAN-RL | PPOD | PPOD with Attention |
|---------|---------------------|--------|------|---------------------|
| Poisson | $0.542 \pm 0.02$ | $0.651 \pm 0.03$ | $0.506 \pm 0.01$ | $0.503 \pm 0.03$ |
| Hawkes | $0.522 \pm 0.01$ | $0.654 \pm 0.02$ | $0.515 \pm 0.04$ | $0.523 \pm 0.08$ |
| MIMIC | $0.592 \pm 0.05$ | $0.601 \pm 0.08$ | $0.593 \pm 0.06$ | $0.582 \pm 0.04$ |
| Taxi | $0.795 \pm 0.01$ | $0.832 \pm 0.02$ | $0.690 \pm 0.01$ | $0.710 \pm 0.01$ |

GAN-RL as the discriminator gets more real sequences and is able to distinguish them better from generated data. However, when $\beta = 0.0$, we notice that GAN-RL gets an AUROC of around $0.5$ suggesting it is purely random.

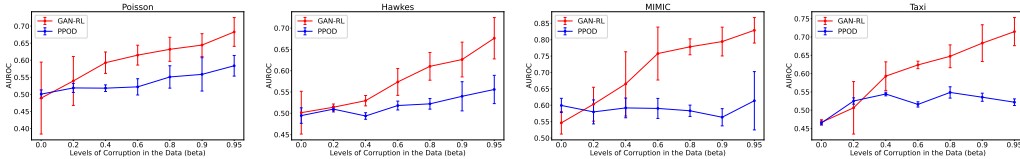

Figure 6: Sensitivity to corruption in the data.

## 4.4 IMPORTANCE OF ATTENTION

The addition of the attention layer was really crucial to the good performance of GAN-RL. We believe this is because the generator needs to be able to focus on particular time points in the past before being able to make decisions on outliers and this is where adding attention can help. We run an ablation study where we remove the attention layer from the encoder and plot the training performance across the last $10\%$ episodes. We do notice a drop in performance of GAN-RLwithout attention in Table 2. It is interesting to note that, although removing attention results in worse performance of GAN-RL in general, the amount of decrease can vary from datasets to datasets. Importantly, even without attention, GAN-RLcan still outperform the baseline across all datasets, especially on Taxi, suggesting there are merits to our overall framework. Meanwhile, just adding an attention layer on top of cLSTM to the baseline does not improve performance much, if at all.

## 5 CONCLUSION

In this work, we developed a novel *unsupervised* event outlier detection framework based on ideas from GANs and RL. RL-based generators are learned to correct outliers in the unlabeled dataset through GAN-based training and then applied to unseen sequences to detect event outliers online. We evaluated our method on both synthetic and real-world datasets with simulated outliers. Compared with the state-of-the-art semi-supervised approaches, our method shows similar or better detection accuracy in all the experiments.

## REPRODUCIBILITY

We describe all the necessary information for the reproduction of results shown in the paper. The encoder architecture is described in Section 3.2 with the detailed hyperparameters to implement the architecture in Table 3 and 4. The generator and the discriminator including the architecture and training is discussed at a high level in Section 3.3 and 3.4. All the hyperparameters required to reproduce the results are in Table 3 and 4. The real-world datasets used also are readily available from their cited papers.

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

# A ALGORITHM DETAILS

## A.1 TRAINING DYNAMICS

At the core of the training methodology for this framework lies the concept of leveraging a single scalar reward function for each sequential dataset. During discriminator training, a generated sequence has a target label 0 and a real sequence 1. In parallel, the RL agent takes on a distinct objective: to enhance the quality of the generated sequences, striving to render them as devoid of outliers as possible, thereby aligning them closely with the characteristics of clean sequences. The incentive guiding the RL agent's actions is grounded in the discriminator's output for classifying sequences as 0. This output is repurposed as a reward signal that propels the RL agent to optimize its approach. This interplay between the RL agent and the discriminator creates a dynamic wherein the generated sequences evolve to closely mimic the clean sequences, blurring the boundaries between the two gradually.

## A.2 TWO TIMESCALE TRAINING FOR OMISSION OUTLIERS

As mentioned in the main paper, we have two agents in an omission outlier detection setting. The second predicts the location of the missing point, whereas the first agent predicts the probability of the presence of a missing point in the given interval. Now, since the credit assignment for the two agents is intertwined, the second agent might be unfairly penalized for mistakes made by the first agent. To prevent, this from happening, we start with a slower learning rate for the second agent to allow the first agent to first learn faster and reduce the negative impact of poor decision-making by the first agent. For our experiments, we double the learning rate every 3000 episode, which starts at $10^{-5}$, 10 times slower than the learning rate of the first agent, $10^{-4}$.

# B DATASETS

## B.1 INTENSITIES FOR SYNTHETIC DATASETS

The synthetic datasets are each characterized by their own intensity functions. For simulating these Point Processes, we use Tick (Bacry et al., 2017). The details are as follows:

- **Poisson**: The intensity function, $\lambda(t) = 1 + \sin(2t)$.
- **Hawkes**: we use a kernel with a sum of $U$ exponential decays with intensities $\alpha = [0.01, 0.02, 0.01]$ and decays $\beta = [1.0, 3.0, 7.0]$. The intensity function is defined as:

$$\lambda(t) = \sum_{u=1}^{U} \alpha_u \beta_u \exp\left(-\beta_u t\right) \mathbb{1}_{t>0}$$

## B.2 OUTLIER SIMULATION

For commissions, we use a Poisson process with a constant intensity function ($\alpha$) to generate outliers, and then we merge them with the clean sequence to create a corrupted sequence. The value of the intensity function is defined based on the type of dataset being used. For omissions, points need to be removed, and the maximum number of points, $P$ to be removed is chosen beforehand. After that, we randomly generate a number $p$ between 1 and $P$, and remove $p$ points randomly chosen from the clean sequence.

As a rule of thumb, we choose $\alpha$ and $P$ such that the number of outliers in a sequence is around $20 - 30\%$ of the average clean sequence length. The exact values for every dataset are in Table 3 & 4.

# C HYPERPARAMETERS

For all the experiments, the hyper-parameters were tuned on final AUROC scores on the training set.

Table 3: Hyperparameters for all the experiments with commission outliers

| Dataset | Generator | Discriminator |
|---------|-----------|---------------|
| Maximum Time Length=10
Poisson: $\alpha$=0.5
Hawkes: $\alpha$=0.5
MIMIC: $\alpha$=0.1
Taxi: $\alpha$=0.3
Seeds:
Training:
$[100, ..., 109]$
Testing: $[1000]$ | cLSTM hidden size=64
Self Attention Layer
Layer Normalization
learning rate for encoder = 0.001

Actor Arch:
Linear(64, 64),
Tanh(),
Linear(64, 64),
Tanh(),
Linear(64, 2),
Softmax()

Critic Arch:
Linear(64, 64),
Tanh(),
Linear(64, 64),
Tanh(),
Linear(64, 2),

update policy every 10 sequences
update policy for 10 epochs in one update
clip parameter for PPO, $\epsilon = 0.2$
$c_1 = 0.5$, $c_2 = 0.01$
discount factor = 0.99
learning rate for actor network = 0.00001
learning rate for critic network = 0.00001 | cLSTM hidden size=64
learning rate for encoder = 0.001

Arch:
Linear(64, 64)
Tanh(),
Linear(64, 1)
Linear Layers have
Spectral Normalization

update discriminator every 50 sequences
learning rate for discriminator = 0.001
Number of Episodes=10000
Update Frequency=1000 |

Table 4: Hyperparameters for all the experiments with omission outliers

| Dataset | Generator | Discriminator |
|---|---|---|
| Maximum Time Length=10
Poisson: $P$=6
Hawkes: $P$=6
MIMIC: $P$=2
Taxi: $P$=30
Seeds:
Training:
$[100, ..., 109]$
Testing: $[1000]$ | cLSTM hidden size=64
Self Attention Layer
Layer Normalization
learning rate for encoder = 0.001

Actor 1 Arch:
Linear(64, 64),
Tanh(),
Linear(64, 64),
Tanh(),
Linear(64, 2),
Softmax()

Actor 2 Arch:
Linear(64, 64),
Tanh(),
Linear(64, 64),
Tanh(),
Linear(64, 1),
Sigmoid()

Critic Arch:
Linear(64, 64),
Tanh(),
Linear(64, 64),
Tanh(),
Linear(64, 2),

update policy every 10 sequences
update policy for 1 epochs in one update
clip parameter for PPO $\epsilon = 0.2$
discount factor = 0.99
$c_1 = 0.5, c_2 = 0.01$
learning rate for actor network 1 = 0.0001
learning rate for actor network 2 = 0.000001
learning rate for actor network 2 doubles
every 3000 episodes till it reaches 0.00001.
learning rate for critic network = 0.0001 | cLSTM hidden size=64
learning rate for encoder = 0.001

Arch:
Linear(64, 64)
Tanh(),
Linear(64, 1)
Linear Layers have
Spectral Normalization

update discriminator every 50 sequences
learning rate for discriminator = 0.001
Number of Episodes=20000
Update Frequency=2000 |

