# OpenReview forum: "Unsupervised Event Outlier Detection in Continuous Time"
_ICLR.cc/2024/Conference — ICLR 2024 Conference Withdrawn Submission_

### Official Review · Reviewer_T8fJ · 2023-10-29

**Soundness:** 1 poor
**Presentation:** 3 good
**Contribution:** 2 fair
**Rating:** 5
**Confidence:** 4

**Summary:**

This paper introduces the GAN-RL method for event outlier detection in continuous time event streams. The method proceeds by learning an encoder for event streams based on an LSTM architecture, which outputs a latent vector for each time point. Then, RL agents are trained to tamper with the TPP by adding and removing point occurrences (thus "generating" a new sequence). Finally, a discriminator is simultaneously trained to identify artificially generated samples and acts as the reward function of the RL agents.

At test time, the RL agents treat each sequence as an episode and score each point and interval as outliers. AUROC of ground truth outliers (omissions and commissions also labelled randomly by authors) are reported.

Overall the paper is well written, sets up and explores the continuous time event outlier detection problem well. However the paper is missing a large ablation study and empirical results to justify the amount of complexity inherent in the methodology is missing.

**Strengths:**

The paper proposes a strong method for event outlier detection in continuous time. Its combination of different ideas from deep reinforcement learning and neural sequence models is novel and interesting, and may be widely applicable in other tasks. It would be interesting to try the ideas in the paper, for example, in time series anomaly detection.

The paper also makes a significant effort to rigorously set up the event outlier detection task where an outlier / anomaly may simply be that an event is "missing." Its treatment of the topic and its survey of related works is insightful and accurate.

**Weaknesses:**

The main weakness of the paper is its lack of ablation studies and reproducibility. The paper introduces an overly complex architecture for a task where naive baselines are often available in practice. The architecture comprises an LSTM for encoding, multiple RL agents (where the policy of one depends on the policy of the other--and not vice versa), a discriminator architecture that could also be seen as the "reward function," design choices in the neural networks in the form of "continuous time" LSTMs, spectral normalization and attention. Only one of these design choices is subjected to an ablation study, where it is seen suprisingly that it is is critical to the success of the method. This invites the question how other complex architectural additions benefit the method and whether they are necessary. In the absence of such a study + open-source code to easily reproduce the work, the proposed method does not appear practical.

Similarly, the paper only compares with the PPOD baseline. However many other naive baselines are missing, such as simply learning a simple TPP model using a PPOD like procedure. Even simpler, see the RND and LEN baselines in the PPOD paper, or baselines in the Shchur et al paper applied on a discretized space, etc. i.e., there appear to be a continuum of ideas from naive baselines to the complex architecture of GAN-RL that need to be explored especially seeing that the task definition is not clear and there is no common task / benchmark for event outlier detection. (i.e., the authors define the task as well as solve it.)

**Questions:**

1. I do not follow the claim that the method is able to "learn from contaminated data" where all other methods assume clean data. I would beg to differ, as all unsupervised outlier detection methods would in practice learn from contaminated data and demonstrate empirically that they are resilient to this contamination. Could you explain this further with references to where else the empirical success of outlier detection methods is conditioned on the availability of clean (i.e., one-class) data and do not work with contaminated (i.e., "two-class," labels not available) data?

---

### Official Review · Reviewer_bmi9 · 2023-10-30

**Soundness:** 3 good
**Presentation:** 2 fair
**Contribution:** 3 good
**Rating:** 3
**Confidence:** 4

**Summary:**

This paper proposes an unsupervised anomaly detection method for identifying unexpected absence or presence of events within an event sequence. The paper uses generative adversarial networks (GANs) and reinforcement learning (RL). The RL component trains a controller that learns the best way to make corrections to the data to add events that should have appeared that did not and remove events that should not have appeared---such events represent anomalies. Also, different from typical GAN-based outlier detection
approaches, the proposed method can detect outliers in an online manner.

**Strengths:**

1. The paper's use of reinforcement learning is clever and is implemented in a way that corresponds well to the intended problem of finding errors of commission and errors of omission together with where the event should appear within the sequence.
2. The paper's study of the impact of the amount of corruption in the data is good---it should be an obvious step that is always taken, but is actually rarely done.

**Weaknesses:**

1. The paper did not evaluate the accuracy of the RL agent that predicts the time at which the omitted event should occur.
2. There are several unsupervised anomaly detection methods for time series against which the authors should compare their algorithm: Examples include:
Das, et. al., Multiple Kernel Learning for Heterogeneous Anomaly Detection, KDD-2010
Melnyk, et. al., Semi-Markov Switching Vector Autoregressive Model-Based Anomaly Detection in Aviation Systems, KDD-2016.
Memarzadeh, M., et. al, “Unsupervised Anomaly Detection in Flight Data Using Convolutional Variational Auto-Encoder,” Aerospace, Vol. 7, 2020.

and a recent semi-supervised anomaly detection algorithm for anomaly detection is:
Memarzadeh, et. al., ”Multi-Class Anomaly Detection in Flight Data Using Semi-Supervised Explainable Deep Learning Model”, Journal of Aerospace Information Systems, 19(2), 2022.


3. Some description of the sensitivity of the algorithm's performance to variations in the hyper parameters needs to be given.

**Questions:**

1. Obviously, I would like to know how the authors will address the items identified as weaknesses.
2. What changes would need to be made to the algorithm to allow for the case where the events may be of different types? The citations that I give in the weaknesses section are intended to address this case.

---

### Official Review · Reviewer_XqpQ · 2023-10-30

**Soundness:** 3 good
**Presentation:** 2 fair
**Contribution:** 2 fair
**Rating:** 5
**Confidence:** 4

**Summary:**

The paper proposes a method for detecting two types of outliers from event sequence data:
- Commission outliers: unexpected occurrences of events
- Omission outliers: unexpected absence of events

Each event sequence is encoded using cLSTM and then self-attention mechanism with a causal mask. For each type of outliers, the encoded sequence is passed to a model consisting of a generator and a discriminator

For commission outliers, the generator classifies each event point as outlier or not, and outliers are removed from the original sequence. The discriminator, having the identical architecture as the generator except for not having self-attention, classifies the modified sequence as real or generated.

For omission outliers, the generator classifies the interval between each pair of consecutive events as missing an event or not, and if yes imputes an event at the identified location. The discriminator works as in the previous case.

After training, the generator can be used to identify if an input event sequence is anomalous.

**Strengths:**

The problem formulation sounds plausible. The proposed network architecture is sensible.

**Weaknesses:**

W1: The claim to be the first paper on outlier detection in an event sequence is somewhat strong. There are already works on the same theme as listed out here:

Manish Gupta, Jing Gao, Charu C. Aggarwal, Jiawei Han: Outlier Detection for Temporal Data: A Survey. IEEE Trans. Knowl. Data Eng. 26(9): 2250-2267 (2014)

Please double check the above paper for potential baselines.

------

W2: The paper should include at least a diagram about the model architecture. This will greatly improve readability.

------

W3: As mentioned in the paper, most sequences passed in the generator and discriminator are normal. This would make the training data very skewed. The discriminator would end up classifying most sequences from the generator as normal. The generator would not receive meaning feedback for refinement from the generator. How should this problem be addressed?

------

W4: For omission outliers, instead of having two RL agents, what about having just one RL agent predicting directly the location of the missing event, and imputing no event if the location is zero? I have concern about cascading error in the present approach.

------

W5: In the experimental results, except for Section 4.1, the other sections lack key takeaways from the results. The authors should improve this.

**Questions:**

Q1: Is there any other relevant baselines?

------

Q2: How would data skewness impact the training process?

------

Q3: How about having just one RL agent for omission outliers?

---

### Official Review · Reviewer_aHPA · 2023-11-06

**Soundness:** 3 good
**Presentation:** 3 good
**Contribution:** 3 good
**Rating:** 6
**Confidence:** 3

**Summary:**

ُThe paper presents an innovative unsupervised outlier detection framework in event sequence data. While the method showcases promising results, addressing the complexity, interpretability, scalability, real-world deployment challenges, and ethical considerations would further enhance the paper's impact and relevance in the field of outlier detection and event sequence analysis.

**Strengths:**

+ The paper's focus on unsupervised methods which is a real setting for event detection.
+ The paper introduces an unsupervised outlier detection in event sequence data. The combination of Generative Adversarial Networks (GANs) and Reinforcement Learning (RL),  to detect abnormal events and unexpected absence of events, is interesting and innovative.

**Weaknesses:**

+ The paper does not discuss the scalability of the proposed method, especially concerning large-scale event sequence data. Considering the potential applications in real-time systems where massive event data streams are common, addressing scalability concerns and optimizing the method for efficient processing of large datasets would be essential.
+ While the paper compares the proposed method with semi-supervised approaches, a broader comparative analysis against various unsupervised outlier detection techniques. Also, the using of point process in event detection is not new and the paper can discuss the related papers, for example, Modeling User Return Time Using Inhomogeneous Poisson Process, ECIR 2019.

**Questions:**

+ Could the authors provide a detailed explanation or an illustrative example of how the GAN and RL components are integrated in the proposed framework?
+ Considering the complexity of the proposed model, are there any specific methods or measures implemented to enhance the interpretability of the outlier detection results?
+ How sensitive is the proposed framework to changes in hyperparameters? Are there any specific hyperparameters that significantly affect the model's performance?
+What challenges do the authors foresee when deploying the proposed framework in real-world applications?